# A Transcriptome Community-and-Module Approach of the Human Mesoconnectome

**DOI:** 10.3390/e23081031

**Published:** 2021-08-11

**Authors:** Omar Paredes, Jhonatan B. López, César Covantes-Osuna, Vladimir Ocegueda-Hernández, Rebeca Romo-Vázquez, J. Alejandro Morales

**Affiliations:** Computer Sciences Department, Exact Sciences and Engineering University Centre, Universidad de Guadalajara, Guadalajara 44430, Mexico; omar.paredes@academicos.udg.mx (O.P.); jhonatan.lopez@alumnos.udg.mx (J.B.L.); cesar.covantes@alumnos.udg.mx (C.C.-O.); vocegued@gmail.com (V.O.-H.)

**Keywords:** mesoconnectome, gene regulatory network, Allen Brain Atlas, graph communities, brain transcriptome, bottom-up modeling

## Abstract

Graph analysis allows exploring transcriptome compartments such as communities and modules for brain mesostructures. In this work, we proposed a bottom-up model of a gene regulatory network to brain-wise connectome workflow. We estimated the gene communities across all brain regions from the Allen Brain Atlas transcriptome database. We selected the communities method to yield the highest number of functional mesostructures in the network hierarchy organization, which allowed us to identify specific brain cell functions (e.g., neuroplasticity, axonogenesis and dendritogenesis communities). With these communities, we built brain-wise region modules that represent the connectome. Our findings match with previously described anatomical and functional brain circuits, such the default mode network and the default visual network, supporting the notion that the brain dynamics that carry out low- and higher-order functions originate from the modular composition of a GRN complex network

## 1. Introduction

The human brain is a complex cell network that yields high-level and complex functions, including memory, reasoning, awareness, and language, among others [1,2,3]. The brain and its ailments are typically explored in several ways. Some methods measure higher-order effects such as electrical activity (Electroencephalography-EEG) and energetic demand (functional Magnetic Resonance Imaging-fMRI). The latter has led to whole-brain cartography with region-wide resolution [4,5,6,7], and to model dynamics in resting-state networks [8,9] and neuropathology, i.e., epilepsy [10,11]. Other studies focus on the molecules, proteins, hormones, neurotransmitters, genes, and transcripts [12,13,14,15].

Transcriptional studies have mapped the brain cartography and described the emergence and development of brain structures, thus pointing to molecular gradients that are likely to drive brain plasticity [16,17,18,19]. Such evidence suggests a pathway between brain transcriptomics and brain connectomics. Richiardi et al. [20] mapped the brain from macro functions to cell populations gradients and gene expressions, setting a top-down transcription approach of the connectome.

Recent works aimed to bridge the gap between the macroscale towards the microscale landscapes demonstrated the genes’ influence on connectome dynamics for *C. elegans* [21], *M. musculus* [22] and even humans [23]. Similar studies screened genes with dimension-reducing methods in previously connected structural or functional brain nodes [24,25]. Their conclusions point to few genes responsible for brain connectivity, yet this does not imply that some scattered genes drive brain structure, for the brain and its functions result from many interacting genes.

The analysis of gene regulatory networks in complex phenotypes has revealed a compartmentalization phenomenon able to aggregate into more complex structures, compounding its effects [26,27,28,29]. It has been hypothesized that a hierarchic organization of these compartments allows genes to reach higher-order phenomena in the human brain, i.e., memory, consciousness, and self-awareness [30].

An approach to exploring those brain compartments is through graph analysis, particularly through the graph mesostructures, such as cliques, communities, and modules. Barabási and Barabási and Kovács et al. studied brain genetics through cliques; however, this work was unable to establish brain-wide connectivity [31,32], while Betzel et al. conducted a comparative analysis of the fMRI voxel communities of various connectomes but was only able to correlate with coexpression networks in the mouse [33].

In contrast, a different, bottom-up approach was proposed [34,35,36], yet experimental works have not been developed as far as we know. In the present article, we explore brain-wise connectivity through functional gene regulatory networks. We identify gene communities related to distinct brain cell phenomena and brain modules that map previously described anatomic and functional brain circuits. Our model is the first attempting to bridge the gap between micro- and macroscale connectomics.

## 2. Materials and Methods

### 2.1. Database and Graph Estimation

We retrieved RNA-seq studies from the Allen Brain Atlas database of the two different donors [37]. Both donors are African-American males, Donor 1 (D1) aged 24 and Donor 2 (D2) 39 years. D1 showed normal neuroanatomy and a history of asthma, while D2 presented a possible small pituitary adenoma. Each RNA-seq consists of 121 post-mortem gene expression segments, corresponding to specific brain regions. To minimize findings from the baseline cellular processes throughout the analysis, we removed transcripts annotated as long non-coding and those related to transcription, translation, and replication (i.e., rRNAs, tRNAs, DNA polymerase, etc.). We estimated the gene expression variance logarithm across all brain regions for each subject according to:(1)$${\hat{s}}_{g_{i}}^{2}=\log\frac{\sum_{n=1}^{N}g_{i,n}-{\bar{g}}_{i}}{N-1}$$
where $g_{i,n}$ correspond to gene expression in a *n* brain region, ${\bar{g}}_{i}$ stands for the mean gene expression through brain, *N* is equal to the 121 studied brain regions, and ${\hat{s}}_{g_{i}}^{2}$ is the gene expression variance.

Low-variance gene expressions were discarded since expression gradient skew high-variance connectivity. To determine the threshold between low-variant and high-variant genes, we performed a sweep of 0.1 increments of the variation logarithm and estimated the genes above the evaluated threshold. Then, we calculated the first and second derivatives of the retained genes number and set the limit of variant genes where both the first derivative is minimum and the second derivative is highest (see Figure 1, on the left).

Jensen–Shannon divergence is an informational distance based on the Mutual Information (Kullback–Leibler Distance) of two probability distributions. The mean probability distribution of both of them is estimated; then their deviation from this distribution is measured, with one representing that both distributions differ while 0 indicates that they are the same. In this work, we aimed to screen those brain gene expressions that had higher similarity as a coexpression biomarker; thus, we chose this measure to build the adjacency matrix.

With the whole-brain expression of high-variant genes ${\hat{g}}_{i}$, we calculated the paired Jensen–Shannon divergence (JSD) (Equation (Equation 2)) among all the genes obtaining a square matrix that reflects the brain expression profiles, $JSD({\hat{g}}_{i}||{\hat{g}}_{j})$. The brain’s gene network was built from this matrix.
(2)$$JSD(g_{\hat{i},n}\left|\right|g_{\hat{j},n})=\frac{1}{2}\sum_{n=1}^{N}g_{\hat{i},n}\left(\log\left(g_{\hat{i},n}\right)-\log\left({\hat{m}}_{i,j,n}\right)\right)+\frac{1}{2}\sum_{n=1}^{N}g_{\hat{j},n}\left(\log\left(g_{\hat{j},n}\right)-\log\left({\hat{m}}_{i,j,n}\right)\right)$$

To retain only the strongest gene-to-gene relationships, we estimated the maximum entropy value of the betweenness distribution according to Monaco’s works [38,39] and selecting the edges with the lowest divergence. The resulting network was from here on treated as an unweighted graph.

### 2.2. Gene Community Analysis (Gene-Wise)

Genes carry out cellular functions in groups; thus, mesoscale structures are ideal for modeling them, i.e., communities, partitions, or clique complex. We first calculated their graph features to analyze those gene communities, i.e., eigenvector centrality, betweenness centrality, closeness centrality, assortativity coefficient, and degree-assortativity metrics. Afterward, we applied the leading-eigenvector method [40]. Which is an eigenspectrum modularity optimization method where the modularity is the quality function:(3)$$Q=\frac{1}{2m}\sum_{i,j}B_{i,j}\delta \left(g_{i},g_{j}\right)$$
where B is the modularity matrix which is estimated by subtracting the connection probability *P* between vertex to the adjacency matrix A, $\delta (g_{i},g_{j})$ is the Kronecker delta function, and *m* is the number of edges. *Q* is optimized by selecting the eigenvalue and the corresponding eigenvector that will maximize it. With the selected eigenvector, we performed spectral clustering, where each cluster corresponds to a community.

A community is a set of genes sharing similar topological properties [41] and potentially co-expressed through the brain. We discarded communities with less than ten genes. For those containing more than 150 genes, we estimated the method once again for each one confirming whether the communities no longer held any other partitions.

We ran 30 times the community stage for each donor and calculated its modularity. To prove that the identified communities are non-trivial features, we compared the donors’ modularity distributions against those of 30 Erdős–Rényi null models using the same community analysis approach.

We enriched the identified communities through ontologies based on the Gene Ontology (GO) with ViSEAGO [42] and FDR corrected with Fisher test p<0.01. We then measured the semantic similarity between the ontologies linked to each community, based on Wang’s method [43] within GOSemSim library [44,45].

### 2.3. Multilayer Analysis (Module-Wise)

Once we obtained the gene communities, we retrieved the individual community’s expression profile for every brain region and calculated the Jensen–Shannon divergence across those regions. Each of these Jensen–Shannon matrices represented brain connectivity subset from each gene community, and the multilayered network represents the whole-brain community-wise connectivity model. We modeled the transcriptional mesoconnectome by binarizing those matrices among zones. We set the threshold for the multilayer links in the corresponding value to the divergence distribution mode for both donors. This minimizes the potential spurious connections while preserving the least divergent through the network layers. We used the term human mesoconnectome for the (macro-level) brain architecture integration derived from gene co-expression circuitries (micro-level).

To analyze the transcriptional mesoconnectome dynamics, we estimated with the MuxViz framework [46] the following multilayer network metrics: degree, page rank, eigenvector centrality, k-core. Degree is the number of links through the layers ignoring the inter-layer link nodes of itself; Page Rank (Equation (Equation 4)) is the probability of a node reaching any node $\frac{\left(1-r\right)}{NL}$, thus ranking the nodes based on the latter probability [47]. As in the single layer, those probabilities are uniform $rT_{j\beta }^{i\alpha }$ through all nodes and interactively updated; however, in the multi-layer case, the probabilities $u_{j\beta }^{i\alpha }$ are passed as the initial values to the subsequent layer [48]. For eigenvector centrality (Equation (Equation 5)), the suprajacency matrix is encoded into an aggregate matrix $M_{j\beta }^{i\alpha }$ via an eigentensor $\Theta_{j\alpha }$. The eigenvector centrality is the dot product of the leading eigenvector $\lambda_{1}^{-1}$ and the neighborhood of each node [48]. The k-core (Equation (Equation 6)) represents the ratio of the coreness $n_{k-core}$ for the probability of specific degree-node $n_{k}\left(q\right)$ through all the layers [49].
(4)$$R_{j\beta }^{i\alpha }=rT_{j\beta }^{i\alpha }+\frac{\left(1-r\right)}{NL}u_{j\beta }^{i\alpha }$$
(5)$$\Theta_{j\beta }=\lambda_{1}^{-1}M_{j\beta }^{i\alpha }\Theta_{i\alpha }$$
(6)$$P_{k}\left(q\right)=\frac{n_{k}\left(q\right)}{n_{k-core}}$$

Modules are clusters of intra- and inter-layer nodes inside the multilayer network. Such modules portray brain ‘circuits’ connected by their shared molecular functions. In this work, we identified these modules following the proposed algorithm in [50] and mapped them to the regions of Allen Brain Atlas.

## 3. Results and Discussion

### 3.1. Data Preprocessing

After removing the annotated transcripts described in Section 2.1 from both donors, D1 and D2 of the Allen Brain Atlas, we collected 22,318 gene expression data of each donor. Monaco et al. suggested that low-variance gene expression is of no relevance to identify target genes associated with brain study conditions [39]. This can also mean that low-divergent genes are ubiquitous, carrying out the most fundamental biological functional architecture. At the same time, those with higher divergence will play a more relevant role in connectome dynamics. To determine the divergent genes, we calculated a threshold based on the first order log-variance resulting in a $log_{10}=4.5$ (see Figure 1, on the left). After setting the threshold, the remaining genes were 7986 for D1 and 7791 for D2.

For an overview of how well defined are variant vs. non-variant genes, we observed their intersection (Figure 2, on the left). While most genes share the common feature (i.e., they are either variant or non-variant in both donors), a small portion are mixed. This is expected, given the intrinsic variability between both donors (e.g., genomics, age, behavior, health, etc.). Moreover, it is common knowledge that genes are pleiotropic, meaning they can produce multiple phenotypic outcomes [51,52] and phenotypic effects between genes have the potential to be cross-linked leading to polygenic functions [53,54]. This has lead to a screening of genes toward identifying those that influence brain dynamics, like the top-down approach of the transcriptional connectome [18,21,23,25]. A different (i.e., bottom-up) approach may study gene functions and orient those functions depending on their gene communities and interactions. Thus, we enriched those expressed genes by retrieving their GOs with an FDR-corrected *p* < 0.01. Figure 2 (on the left) shows a Venn diagram of their respective GOs (see Appendix A). Except for two ontologies, those functions are either present brain-wise or vary within regions; note that variant sets share most functions. While the former indicates that some functions are more related to general brain tissue architecture, the latter suggests interacting gene communities whose functions perform more local endeavors.

### 3.2. Gene-Wise Analysis

To understand the gene interactions among brain regions, we built similarity matrices among all genes based on their Jensen Shannon divergence for each donor then assembled gene regulatory adjacency matrices from the binarized divergence values below the 41st percentile for D1 and 45th percentile for D2, i.e., 0.118964 and 0.111294, respectively. These values were set following Monaco’s criterion [38] where the threshold is the percentile with maximum entropy value of the betweenness distribution (Figure 1, on the right).

We filtered out all zero-degree subgraphs from both gene regulatory adjacency matrices. The two resulting gene regulatory networks (GRNs) included 7914 nodes and 13,070,196 edges for D1 and 7730 genes/nodes and 13,653,610 edges for D2. We analyzed the GRNs with the metrics described in Section 2.2, and displayed in Figure 3 corresponding to both donors. The figure shows a positive correlation between Degree and Eigenvector centrality (EC), showing that genes with higher degree provide an easy pathway to connect all genes. In comparison, those of lower degree are more distant from them, which suggests a core-periphery trend of the GRN. Such feature is usually related to a high assortativity. The assortativity coefficient resulted in 0.248 for D1 and 0.234 for D2. All the previous results plus degree-closeness high correlation and degree-betweenness low correlation suggest that the brain GRN is a distributed network [55,56,57] with a tendency to node dissociation, meaning that nodes of the same degree do not necessarily connect each other.

Results of D1 and D2 degree-assortativity show a known three-phase behavior [58] (see Appendix A): Phase I, an initial dissortative trend, where high and low degree nodes tend to link; Phase II, an assortative trend, where same-degree nodes attach together; and, Phase III, a final neutral trend, where nodes connect among them no matter its degree [59]. Such complex feature is associated with networks that have high modular behavior and a robust information flow [60], essential traits for connectome GRNs [18,23,24,25].

Most connectome studies focus on macro-level brain modeling employing observable dynamics on functional connectivity [61,62,63,64] followed by the search of select genes that show variation inside their macroconnectome [21,22,25,65], suggesting that only a fraction of genes drives the brain’s identity and its functionality. Such top-down modeling drives to a narrow gene screening, constraining the brain endophenotype spectrum which makes it unlikely to find any brain’s transcriptional network responsible for the brain’s wiring and functions. Meta-analysis of brain genes notes this limitation and discuses the need to extend the gene pool that is evaluated in a brain-wise approach [66,67]. Some bottom-up studies of GRNs across different tissues and diseases have shown how distributed modulation across a wide range of genes plays essential roles in function and pathophysiology [68,69,70,71]. Here we present a bottom-up model [34] that explores the molecular assemblies (in terms of gene expression) and the gene communities as the dynamic clusters that constitute functional regulatory units, as a first step to understand gene regulatory networks as the precursors of a brain archetype.

In this work, we set up a GRN community detection workflow. By choosing communities instead of cliques and rich clubs, we obtained the highest number of mesostructures in the network hierarchy organization while ensuring lesser topological constraints [33,72]. A clique analysis would only retain minuscule fully connected gene subgraphs, restricting the identification of specialized brain pathways [31,32]. In contrast, a rich-club analysis would yield essential genes that modulate the GRN [73,74] without considering the complexity of transcriptional mechanisms to carry out the basal molecular pathways within the brain.

Community detection through the leading eigenvector method yields variable results due to the singular value decomposition algorithm used to estimate each eigenvector. Thus, we repeated our detection 30 times for each donor and measured the variability of the gene composition for every resulting community. Variability gene distribution between communities behaved as a log distribution, inflecting at 10-gene communities. After removing all communities with less than 10 genes, six communities remained for D1 with communities between 753 and 1841 genes, while the seven remaining communities for D2 had a range between 862 and 1893 genes, as shown in Table 1 (gene lists for each community in Appendix A). To test if such communities were not the result of chance [75,76], we compared our data’s modularity distribution with 30 degree-null modularity distribution models. We obtained significantly higher modularity for both partitions (ANOVA $p<2.20\times 10^{-196}$ and $p<2.33\times 10^{-192}$ for D1 and D2, respectively) against the null models’ partitions.

We enriched the identified communities based on the Gene Ontology (GO) Consortium annotation. This yielded a high number of GOs associated with neuronal processes along with various cognitive processes. This was expected since the mapped communities were assumed to provide a transcriptional identity of the brain’s functions. Representative GOs for analysis are highlighted for biological processes in Figure 4 and for cellular components and molecular functions in Figure 5, whereas all GOs and their significance level may be found in the Appendix A.

Next, we measured the GOs semantic similarity among communities, considering both inter- and intra-subject analyses (see Appendix A). On intra-subject examination, we observed that communities are semantically different, suggesting functional compartmentalization. For example, the D2_C3 community is a particular case since it is highly similar to other communities. This may be explained due to the applied method that compares the ontological hierarchy considering all ancestor terms [43] bias a high similarity between communities if a child GO is involved. With these results, we searched for specific functions in each community.

To summarize and insight into the biological role of each community, we built word clouds with their GOs (see Appendix A). For donor 1, we identified two communities associated with general neuronal activity phenomena, D1_C5 immunological activity in neurons and D1_C6 to signaling and brain activity. Community D1_C3 is involved in neuronal cell proliferation and differentiation [77,78,79], where differentiation is driven by neurogenesis kinase-mediated pathways [80,81].

The remaining three D1 communities are related to neuronal activity. These include mechanisms involved in cognitive dynamics, such as the D1_C1 community associated with the neuronal lipidome and key pathways for cognitive functioning [82,83], as well as the D1_C2 community linked to the regulation of neuronal cell lifespan through potentially cell trimming [84]. D1_C4 is involved in axonogenesis, which, together with the former communities, potentially regulates brain plasticity [85,86].

For Donor 2, we encountered communities linked to neuronal processes, albeit general. Such is the case of the D2_C3 community associated with dendritic spine regulation [87] and D2_C2 with a role in neuroimmune modulation that altogether might influence brain repair [88] or circadian cycles [89]. Equivalent to the D1_C6 community, the D2_C1 community is involved in signaling.

We also identified two communities linked to specialized neuronal functions, D2_C5 counterpart to the D1_C4 associated to axonogenesis, while D2_C6 controls and regulates the dendritic tree or dendritogenesis. Both mechanisms, alongside spinogenesis, have a crucial effect on neuronal development and proliferation [90,91,92]. We outlined community D2_C4 involved in angiogenesis. Adult brain angiogenesis is typically considered halted or null, except for pathological processes such as ischemia and tumors [93,94]. Yet, together with their aforementioned neighboring communities, they can be summarized as neurotrophic phenomena. Lastly, the D2_C7 community dedicates to processes involving the choroid plexus as well as cerebral ventricles. The interpretation is that this community is associated with the blood–brain barrier and the cerebrospinal fluid system [95,96].

Two factors previously reported in [25] may explain the low degree of correspondence between D1 and D2 communities. First, donor 2 has shown the widest transcriptome variability of all six donors in the Allen Human Brain Atlas. Furthermore, second, since donor 2 had a possible pituitary adenoma, this may bias community topologies and thus their ontologies [97].

### 3.3. Module-Wise Analysis

The community analysis represents a perspective of performing functions throughout the brain. Yet, all communities at each brain region operate simultaneously, and a single-graph approach oversimplified the transcriptional dynamics. To start a mesoconnectome approach, we selected each community across all brain regions then created a new connectivity graph where the edges are weighted according to their expression profile similarity for that community. Such a connectivity graph represents a layer of an *n*-layered network, where *n* corresponds to each identified community.

To our knowledge, no previous work has reported multilayer network models based on genes. Thus, a reference threshold has not been proposed. In the present study, we decided to set as a lower-bound the percentile corresponding to the distribution mode in both donors. These percentiles are 23 and 27 for donor 1 and donor 2, respectively. We discarded the central measures (i.e., average and median) as threshold candidates due to a highly biased brain region connectivity degree that showed only two modules with modularity (around 0.2).

We estimated the metrics described in the Methodology (see Section 2.3) for multilayered networks. Figure 6 shows their distributions and tendencies. Note a positive correlation between degree and page rank (PR), an expected trend given that nodes the more connected, the more popular are in a graph. There is a positive relationship between degree and eigenvector, reinforcing the importance of the most connected brain regions since their neighbors are interconnected as well. Yet, no clear tendency is evident for PR and eigenvector centrality, which supports the notion of not single but several genes expressing together across brain regions.

The idea of connectome modular topology is widely described in the literature, although largely approached from the macro- perspective [1,2,3]. We found a logarithmic correlation between k-core and eigenvector, with an apparent convergence to a k-core value (72 and 86 for D1 and D2, respectively), something previously suggested as typical in multilayer networks caused by the occurrence of efficient information distributor nodes within one layer that is prevalent between layers [98] in the mesoconnectome. Furthermore, the pronounced presence of k-cores leads towards a modular topology of the mesoconnectome, a feature that has been reported in [99]. Observing such behavior on a bottom-up model based on brain transcriptomics provides support for a multi-scale brain model [36].

To outline the connectome, we evaluate how the different brain areas communicate between them through gene communities. The rationale for this is that intra-community expression modulation works as a communication channel; under this assumption, we modeled the connectome as a multi-layer network, where each layer is the modulation degree among the communities via brain regions. Then, to explore the network dynamics, we estimate various graph metrics showing a tendency to work as integrative brain modules. To outline the modules that compose the mesoconnectome, we carried out a Louvaine community analysis, which is the most widely used method for multilayer graphs [100,101].

For donor 1 we found 13 modules whereas for donor 2 there were 9 (see Appendix A). Such modules are color-coded on the multigraph for both donors in Figure 7. For multilayer module decomposition, it is important to evaluate each layer’s modularity to ensure that each partition properly integrates the layers. These results appear in Table 2. Given the resulting modularities, we can assert that the partitioning is optimum [102].

Figure 8 shows the brain region connectivity through the genes they shared in all brain modules (Full names of brain regions in Appendix A).

The superior frontal gyrus was the most interconnected structure according to our model. Its high connectivity with additional brain structures has been described both anatomically [103] and functionally [104], alongside with them, they yield higher-order cognitive functions. Other high-connected region was the angular gyrus, which is reported as a hub in semantic [105,106] and musical [107] processing networks. Both gyri have been linked to the social brain, comprising networks for memory processing, communication, attention and more [108]. Thus, supporting their high level of connectivity.

The modules found within the mesoconnectome identify some anatomic-functional networks previously described in the literature. The best example is the cerebellum brain structure shown with edges and areas in purple in Figure 8. These brain regions for D1 are wired by module 7, while for D2, no single module links all these regions. For D1, the brain module is built by the D1_C2 and D1_C6 communities, whose roles are neuronal lifespan regulation and brain activity signaling. Altogether, such communities are likely to integrate an area highly intra-connected and specialized, like the cerebellum [109,110,111].

Functional macroconnectivity studies have uncovered distinct integrated anatomic-functional brain networks, for example, the default mode network (DMN) observed during test-resting states [112,113], or the default visual network (DVN) involved in visuospatial functions [114,115]. Thus, we scanned for any modules resembling the networks as mentioned earlier.

For the DMN, we identified three modules that mapped to such network in D1, modules 1, 4, and 6 (orange links in Figure 8). They also link other brain regions previously labeled as gateways to different brain networks, for example, the insular gyri for the executive network [116], and consciousness-related networks [117], and the parietal lobe for social interaction networks [118]. For D2, the DMN mapped modules are fuzzier, but their approximate counterparts are modules 2, 3, 4, and 5.

Modules that integrate the DVN are 3 and 4 for D1, whereas in D2, no equivalent modules were identified, thus resulting from a higher module cohesion compared to D1, which likely leads to intersecting modules with DMN ones. There are other regions included in the DVN modules with visual functions, such as the fusiform gyrus [119,120] and the temporal gyri [121,122], as well as sensory integrative regions as the insula [123,124], and superior parietal lobe [125,126].

Both network cases unveil a piece of complex transcriptional machinery behind the anatomic-functional brain circuitry since these circuits are rendered from multiple metabolic pathways and interactions. This hints at a bottom-up molecular integrative drift towards brain structures with cognitive capability, as proposed by van den Heuvel et al. [36].

Another case to highlight is module 2 in donor 1, which contains the regions of lingual gyrus, angular gyrus, middle temporal gyrus, fusiform gyrus, and putamen. Such mesoconnectome module may be related to language processing [127], since these areas are reported to participate alone or coupled with some of them in visual signs processing phenomena as well as in semantic processing [128]. This module is potentially comparable to module 4 of D2. The findings of our study are not only supported by analyzing the brain zones and their associations reported in the literature. The case of Foxp1 and Foxp4 genes, recognized as genes associated with language functions [129]. Here we observe that they belong to D1-C2 and D1-C3 communities for D1, respectively. Such communities within the modules connect brain structures mainly associated with language processing (i.e., lingual gyrus, hippocampal gyrus).

One last exciting example is module 7 of donor 1, a cluster of areas that include the amygdala-related regions: frontal gyrus (inferior, middle, and superior), angular gyrus, and supra angular gyrus whose functions cover speech processing [127], attentional processes [130], and self-awareness [131]. Altogether, these are intrinsic higher-order brain functions implicated in consciousness ([132], Part 6).

Nonetheless, this module was missing in donor 2.

The present work is exploratory, and we were able to report some definite examples of mesoconnectome circuits. Still, our results are limited by the dismally small brain expression data set, which leads to low statistical power to characterize the inter-subject modules within the gene regulatory network that build the mesoconnectome.

## 4. Conclusions

In this article, we present a bottom-up transcriptome modeling of brain connectivity that led to the identification of brain circuits previously described in the literature. Our results support the notion that the brain dynamics that carry out low- and higher-order functions originate from the modular composition of a GRN complex network.

## Figures and Tables

**Figure 1 entropy-23-01031-f001:**
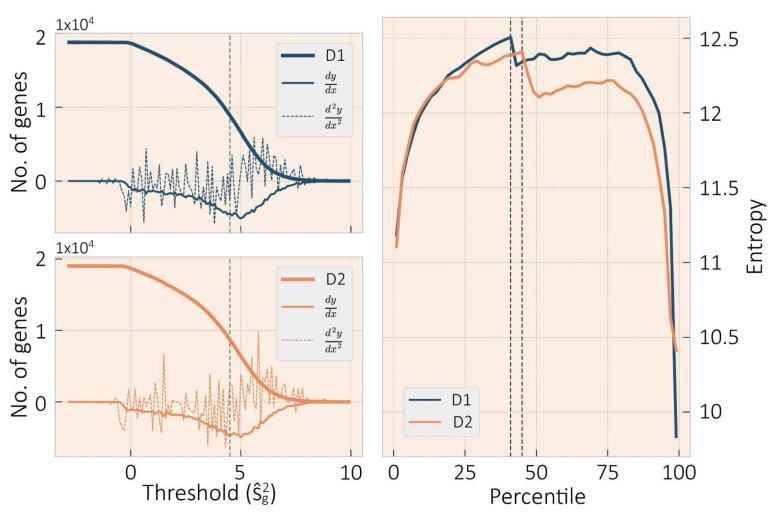
The **left** plots displays the retrieve genes curve for D1 at the **top** and D2 at the **bottom**. The threshold was set on gene expression variance 10-log equal to 4.5 to determine variant genes; the first and second derivatives for both donors are displayed as well. On the **right**, the betweenness entropy to determine the effective edges according to Monaco approach [38], dotted line for the edge threshold, for D1 at the 41st percentile, and D2 at the 45st percentile.

**Figure 2 entropy-23-01031-f002:**
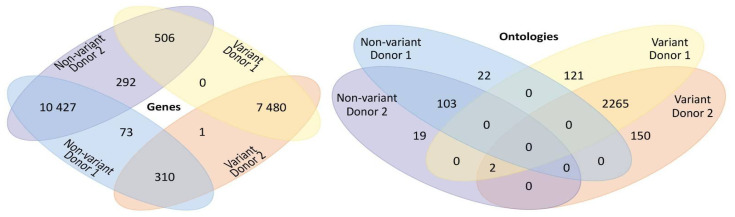
Venn diagrams of Genes variant and invariant (on the **left**) and their GOs entries (on the **right**) for both D1 and D2. Note a marked division between the GOs of variant and invariant genes, pointing to different biological roles of both gene sets.

**Figure 3 entropy-23-01031-f003:**
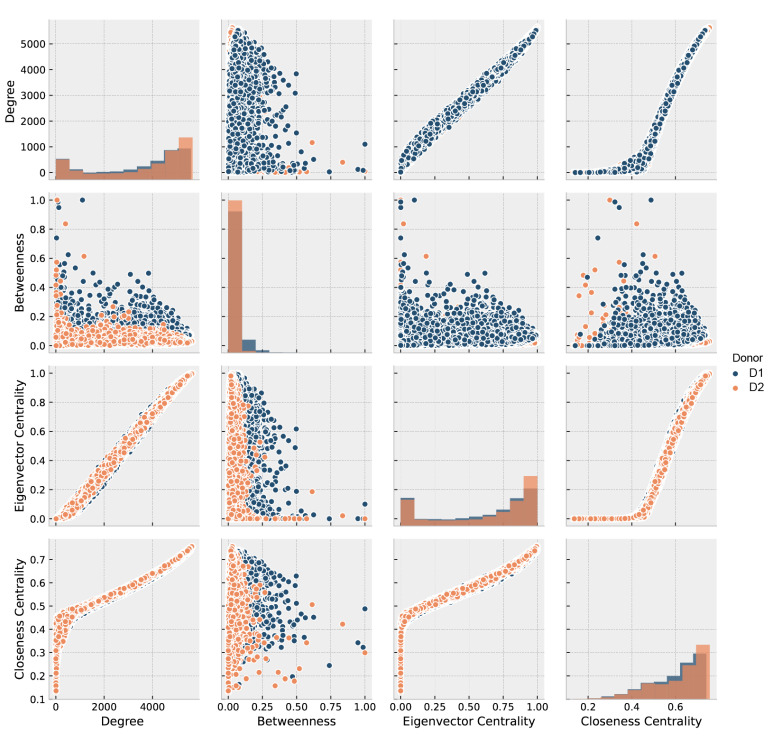
Dispersion plot of GRN metrics: degree, betweenness, eigenvector, and closeness centrality, in dark blue for D1 and orange for D2. In the principal diagonal, distribution of each metric.

**Figure 4 entropy-23-01031-f004:**
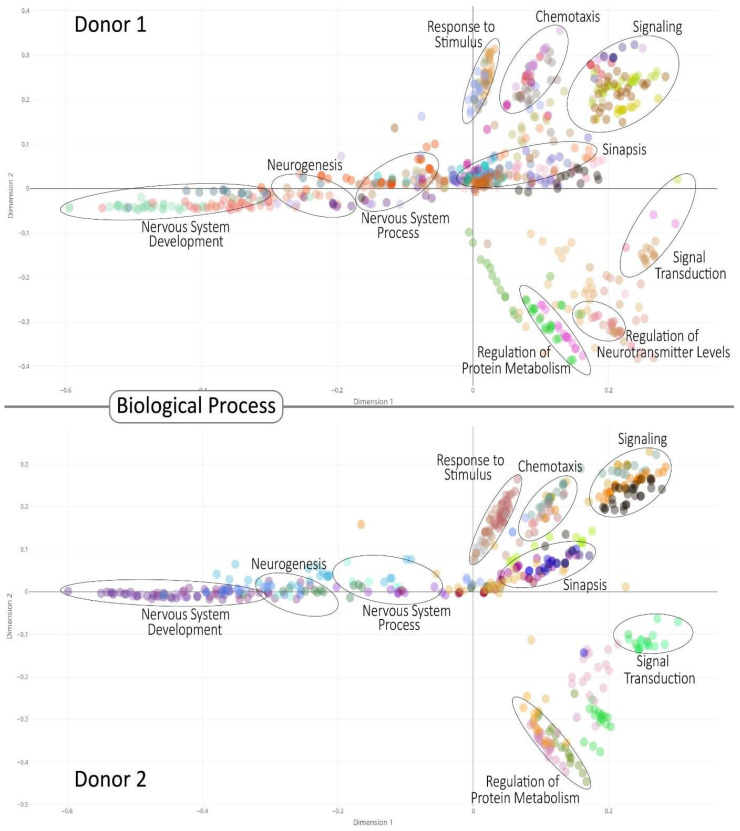
Multi-Dimensional Scale plot of biological process GOs for all donor. The color represents clusters of related functions. Circled are examples of specific brain-related functions. The interactive figures of these GOs for each community and their corresponding *p*-values are Appendix A. Colors between donors are self-generated.

**Figure 5 entropy-23-01031-f005:**
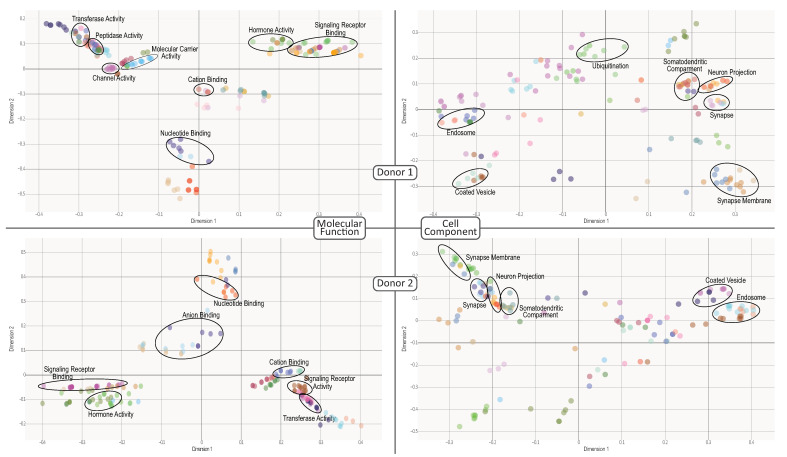
Multi-Dimensional Scale plot of the molecular function and cell component GOs for all donor. The color represents clusters of related functions. Circled are examples of specific brain-related functions. The interactive figures of these GOs for each community and their corresponding *p*-values are Appendix A. Colors between donors are self-generated.

**Figure 6 entropy-23-01031-f006:**
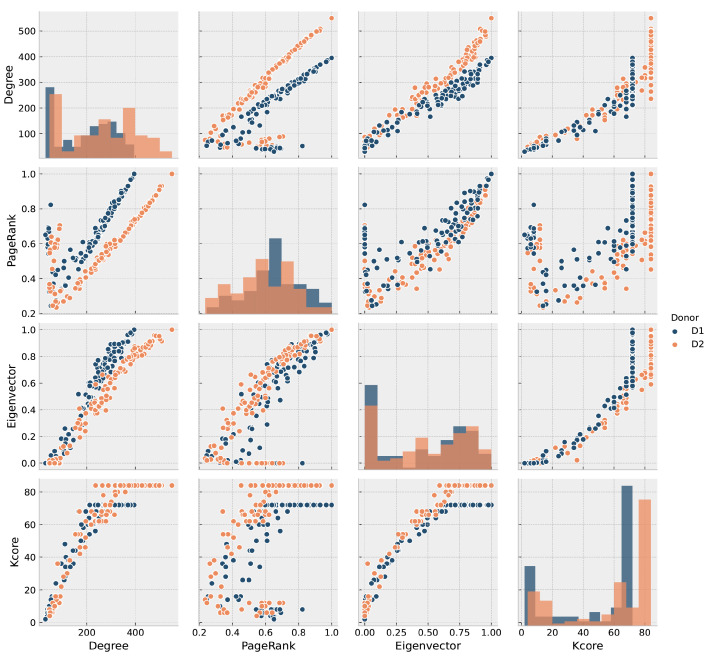
Dispersion plot of multilayer mesoconnectome metrics: degree, PageRank, eigenvector, and k-core. Dark blue is for D1 and orange for D2. In the principal diagonal, distribution of each metric.

**Figure 7 entropy-23-01031-f007:**
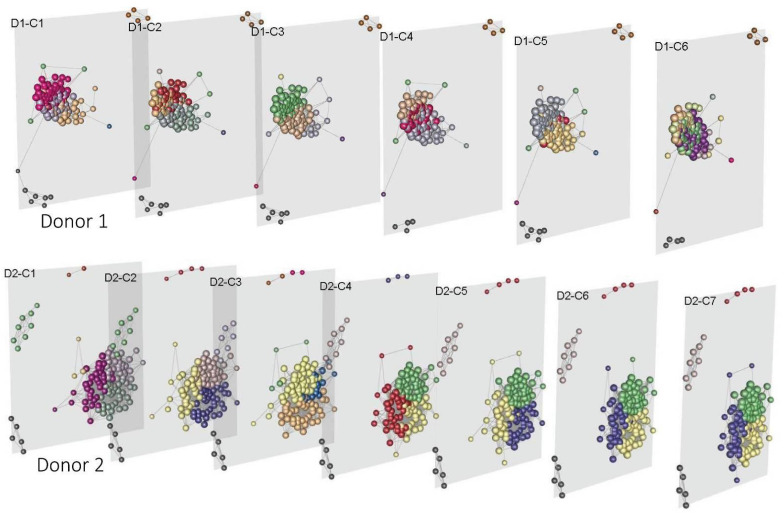
Human mesoconnectome, each layer maps brain connectivity through the gene communities. Colors represent the brain modules that integrate it.

**Figure 8 entropy-23-01031-f008:**
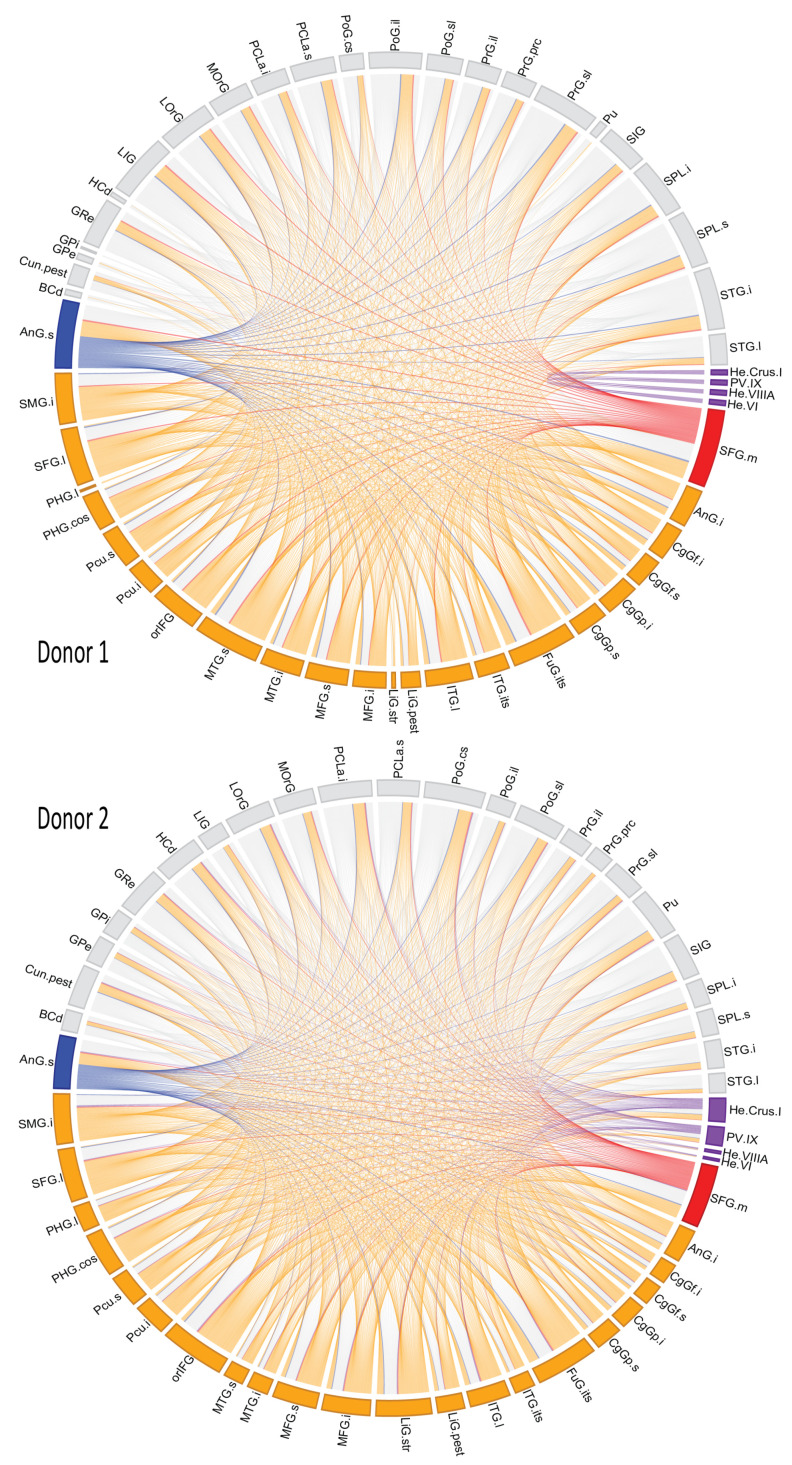
Allen brain regions connectivity is based on shared genes within the brain modules. In purple are the brain regions related to the cerebellum; in orange, DMN brain regions. Red and blue are the densest connected brain regions, the medial bank of superior frontal gyrus (SFG.m) and the superior bank of the angular gyrus (AnG.s), respectively.

**Table 1 entropy-23-01031-t001:** Communities for D1 and D2 and their gene counts.

Donor	Community	# Genes	Donor	Community	# Genes
D1	D1-C1	1138	D2	D2-C1	889
	D1-C2	1356		D2-C2	1127
	D1-C3	1830		D2-C3	862
	D1-C4	1841		D2-C4	872
	D1-C5	753		D2-C5	1893
	D1-C6	941		D2-C6	887
				D2-C7	1141

**Table 2 entropy-23-01031-t002:** Modularity for each layer is estimated based on their brain module partition.

Layer	Modularity	
	D1	D2
1	0.241	0.222
2	0.229	0.307
3	0.18	0.341
4	0.222	0.178
5	0.193	0.236
6	0.34	0.261
7		0.241

## Data Availability

Not applicable.

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
