# Peer review of "A Transcriptome Community-and-Module Approach of the Human Mesoconnectome"

_entropy, 2021, doi:10.3390/e23081031_

Round 1
Reviewer 1 Report
A transcriptome community-and-module approach of the human mesoconnectome.
Using the bottom-up modeling of a gene regulatory network, the authors provided interesting information for gene communities related to distinct brain cell phenomena and brain modules that map to previously described anatomic and functional brain circuits. The authors provide interesting points of view about the brain dynamics that originate from the modular composition of a gene regulatory network.
Yet, there are several points that should be addressed by the author. I would recommend the manuscript for publications after suggested revision.
1. Page 2, line 53-54; Page 4, line 128;.:
Individual properties of the two selected donors, D1 and D2, are quite important for understanding the obtained results. Please describe in the MS.
2. Page 2, line 71.
The "Jensen-Shannon divergence" is not well known to the readers other than information technology. It would be better to briefly describe "Jensen-Shannon divergence" and obtained network properties.
3. Page 3, line 1-122.
In the MS, important terms, "mesoconnectome", "community", "module", and "layer" are used.
However, description is not clear and is very confusing in the Materials and Methods as well as Results section. The authors need to provide a detailed definition for these terms and clarify the meanings.
4. Page 4, Figure 1 legend.:
The description for the sub-figures do not coincide with those in the Figure 2 (Figure 1, On the left, On the right; Figure 2, (a), (b)). Please fix this.
5. Page 7, line 193.: "patophysiology" should read "pathophysiology".
6. Page 7, line 210-213.:
Since the member genes in the individual community is worth noting, please provide the gene lists of each community as a supplementary Table.
7. Page 7, line 221-222.:
Figure 4 and Figure 5 show GOs for the identified communities, but intended community is not clear. Are these GOs for the individual community or all the communities. Please precisely describe in the main text and Figure legends.
8. There are some ambiguous wordings and use of grammar. I suggest to have some external language editing done by a person familiar with the field.
Reviewer 2 Report
The authors use graph theory to explore the gene regulatory networks of the human brain on mesoscale. They estimate the gene communities across all brain regions from the Allen Brain Atlas transcriptome database, which allows them to identify communities for specific brain cell functions, on which they apply some graph theoretical metrics. All the analysis seem quite thorough, and the bottom-up approach is well described, but the link to the brain regions structure and function is still not so clear. In this sense the conclusion should be extended, especially in framing the results from the large-scale brain network modeling and connectomics perspecives. To summarize, the work is important though authors seem to fall bit short of the promises to match those communities with previously described anatomical and functional brain circuits.
Below are my more concrete comments:
- when introducing the importance of the brain as a network, and the methods that measure EEG and fMRI, I think for the findings that the authors give it would be of even more significant to mention the large-scale modelling efforts that try to link brain structure with function with causal models that mechanistically link brain structural features with model parameters. Besides brain connectivity, these rely more and more on regional variance of data which can be important for the brain at rest Kringelbach et al PNAS 2020, but also in diseases such as epilepsy Jirsa et al Neuroimage 2017, Courtiol et al J Neurosci 2020. Another aspect of the structure that shapes the brain dynamics is the spatio-temporal organization of the white matter tracts, which defines the synchornization of the brain, e.g. Petkoski et al PRTSA 2019.
- I find Fig.8 to be most important, at least for linking the work to the large scale computational neuroscience and connectomics, so I would welcome any extension of this results. So for example, it would be interesting the results from Fig.8 to be also shown in regard to Resting State Networks of the brain.
For example authors say: “Other example is the module 2 for D1 that cluster the lingual gyrus (striate and
335 peristrate), peristrate cuneus, precuneus gyrus and parahyppocampal gyrus, brain zones
336 that are associated to visual processing [104] and memory encoding [105–107]. This
337 brain circuit is slightly similar to module 1 in D2.”
This suggest that there might be high similarity between the module 2 and the visual RSN. Also earlier the parahipocampal gurus is discussed as a part of the DMN, which again makes it very tempting to look at which module has the highest similarity with DMN.
- similarly, maybe the authors can make a links between the modules and anatomical structures of the brain such as lobes of the cortex, or the anterior/posterior gradient of the regions.
- can the authors comment (and quantify) based on the literature how much the reported results are expected to be subject specific?
Minor comments:
- fig. 2 seems bit too large? and also Figs. 4-5
- region acronyms in fig.8 are barely visible.
- is the sentence “…which allowed us to identify specific brain cell functions (e.g. neuroplasticity, GABAergic and glutamatergic communities).” correct?
Round 2
Reviewer 2 Report
Following authors revisions, I suggest the manuscript to be accepted for publication.